# Leukocyte Telomere Length as a Marker of Chronic Complications in Type 2 Diabetes Patients: A Risk Assessment Study

**DOI:** 10.3390/ijms26010290

**Published:** 2024-12-31

**Authors:** Krzysztof Sawicki, Magdalena Matysiak-Kucharek, Daria Gorczyca-Siudak, Marcin Kruszewski, Jacek Kurzepa, Lucyna Kapka-Skrzypczak, Piotr Dziemidok

**Affiliations:** 1Department of Molecular Biology and Translational Research, Institute of Rural Health, 20-090 Lublin, Poland; matysiak.magdalena@imw.lublin.pl (M.M.-K.); marcin.kruszewski@gmail.com (M.K.); kapka.lucyna@imw.lublin.pl (L.K.-S.); 2Department of Diabetes, Institute of Rural Health, 20-090 Lublin, Poland; daria.gorczyca@gmail.com (D.G.-S.); dziemidok.piotr@imw.lublin.pl (P.D.); 3Centre for Radiobiology and Biological Dosimetry, Institute of Nuclear Chemistry and Technology, 03-195 Warsaw, Poland; 4Department of Medical Chemistry, Medical University of Lublin, 20-093 Lublin, Poland; jacek.kurzepa@umlub.pl; 5World Institute for Family Health, Calisia University, 62-800 Kalisz, Poland

**Keywords:** type 2 diabetes, telomere length shortening, diabetic chronic complication

## Abstract

Telomere shortening has been linked to type 2 diabetes (T2D) and its complications. This study aims to determine whether leukocyte telomere length (LTL) could be a useful marker in predicting the onset of complications in patients suffering from T2D. Enrolled study subjects were 147 T2D patients. LTL was measured using a quantitative PCR method. Key subject’s demographics and other clinical characteristics were also included. T2D patients with the shortest LTL had higher TC and non-HDL levels, compared to subjects with the longest LTL (*p* = 0.013). Also, T2D patients suffering from diabetic nephropathy showed significant differences in LDL levels (*p* = 0.023). While in the group of T2D patients with diabetic retinopathy, significant differences were observed for parameters, such as duration of diabetes (*p* = 0.043), HbA1c (*p* = 0.041), TC (*p* = 0.003), LDL (*p* = 0.015), Non-HDL (*p* = 0.004) and TG (*p* = 0.045). Logistic regression analysis confirmed a significant risk of association of TC and Non-HDL levels with LTL in the 3rd tertile LTL for the crude model adjusted for sex and age, with respective odds ratios of 0.71 (95% CI 0.56–0.91) and 0.73 (95% CI 0.58–0.91). No significant associations were found between LTL in T2D patients and the prevalence of common T2D complications. Nevertheless, a significant association was demonstrated between LTL and some markers of dyslipidemia, including in T2D patients with either diabetic nephropathy or retinopathy. Therefore, analysis of LTL in T2D patients’ leukocytes demonstrates a promising potential as a marker in predicting the onset of complications in T2D. This could also help in establishing an effective treatment strategy or even prevent and delay the onset of these severe complications.

## 1. Introduction

Diabetes poses one of the biggest global challenges for public health. Type 2 diabetes (T2D) accounts for approximately 90% of all diagnosed cases of diabetes [1]. It is thus estimated that there are currently about 415 million people worldwide suffering from T2D. Moreover, it is predicted that around 580 million people will have this condition by 2030 [2,3]. The overall burden of disease is assessed using the disability-adjusted life year (DALY), a time-based measure that combines years of life lost due to premature mortality (YLLs) and years of life lost due to time lived in states of less than full health, or years of healthy life lost due to disability (YLDs). One DALY represents the loss of the equivalent of one year of full health. In 2021, 58.9 million DALYs or 76.5% of DALYs due to T2D were attributable to risk factors. Of the 16 risk factors high body mass index (BMI) was the primary risk factor for T2D worldwide, accounting for 52.2% of global T2D DALYs [4].

T2D is an age-related disease influenced by both genetic and environmental factors [5]. The condition is characterized by insulin resistance, hyperglycemia, dyslipidemia and other metabolic disorders [6]. Although T2D is a progressive disease, its progression varies widely among patients and depends on their individual characteristics [7]. Along with development of the disease, T2D leads to numerous structural and functional changes occurring in patients’ blood vessels [8]. The most important microvascular complications of T2D include diabetic retinopathy (DR), diabetic polyneuropathy (DPN), diabetic foot ulcer (DFU), and diabetic nephropathy (DN) [9]. DR is a common complication of diabetes and is a major cause of blindness in the working-age population [10]. About one-third of people with T2D show symptoms of DR, and the prevalence increases with duration of T2D [11,12]. DPN can occur in up to half of adults with T2D [13,14] and has been shown to affect mostly nerves of the limbs, leading to various forms of sensory dysfunction [15]. A direct result of progressive DPN of the lower extremities is the onset of DFU, which is estimated to develop in about 25% of patients with T2D. This complication is characterized by multiple ulcerations and soft tissue destruction. It is also associated with the onset of gangrene and an increased risk of amputation [16,17]. DN occurs in about one-third of patients with T2D and is associated with loss of kidney function with increased fibrosis, thereby leading to end-stage renal failure [18].

Standard therapy and management regimens for T2D patients mainly include stabilizing blood glucose, blood pressure, and serum cholesterol levels. Depending on the incidence of additional complications in T2D patients, dedicated treatment protocols are used. For DR, most patients may require frequent anti-vascular endothelial growth factor (anti-VEGF) intravitreal injections, and for patients with proliferative DR, panretinal laser photocoagulation has proven effective in reducing the risk of vision loss [19]. In the case of DPN treatment, various medications are generally used to alleviate symptoms (including chronic pain) and therapeutic drugs targeting the pathogenesis. Non-pharmacological therapies are also used to supplement drug treatment. These include psychological support, acupuncture, physiotherapy and transcutaneous electrical nerve or muscle stimulation [20]. DFU therapy mainly focuses on treatments including surgical debridement of the wound, reducing the pressure exerted by body weight on the ulcer, and treating lower extremity ischemia and infection of the foot. In addition, therapy may include taking medications to accelerate wound healing and targeted oral antibiotics for localized osteomyelitis [21]. The therapeutic scheme for DN includes tight glycemic control, blood pressure control with renin-angiotensin-aldosterone system (RAAS) inhibitors, lipid-lowering agents, weight loss, and protein restriction. In addition, prevention of DN progression uses vitamin D receptor activators, incretin-related drugs, and inflammation-targeted therapies [22].

Telomeres are nucleoproteins containing tandem repeats of *TTAGGG*, 9–15 kb in length at the ends of eukaryotic chromosomes. The main function of telomeres is to stabilize the genome and protect against cellular ageing and apoptosis [5,6]. Telomeres naturally shorten after each cell cycle. Subsequently, cells with critically short telomeres undergo so-called replicative senescence, usually ending with apoptosis. A critical shortage of telomere lengths is referred to as the so-called HayFlick limit [23]. Studies have shown that shortened telomere length correlates with a number of age-related conditions, such as cardiovascular disease, neurodegenerative disease, cancer and metabolic syndrome [24,25,26]. Telomere shortening has been shown to be induced by, amongst other factors, excessive oxidative stress and inflammatory processes, involving mediators such as TNF-α, IFN-γ and IL-6 [26,27] (Figure 1). Furthermore, it has been observed that telomere DNA is sensitive to oxidative damage resulting from free radical exposure [28]. It has therefore been suggested that shortened telomeres may serve as a useful marker of biological ageing [6,29]. The phenomenon of telomere shortening has also been linked to T2D, as well as its complications [25,30]. It has been pointed out that the correlation between telomere length and the incidence of diabetes may be influenced by type of diabetes, gender, age, and BMI [2].

The purpose of this study is to determine if leukocyte telomere length (LTL) from T2D patients can be a useful marker in predicting the onset of complications. A secondary aim of this study was to find any correlation between LTL and patients’ biochemical blood parameters that can be useful in prediction of the onset of disease. This could also help in establishing an effective treatment strategy or even prevent and delay the onset of these severe complications.

## 2. Results

The studied cohort consisted of 147 patients diagnosed with T2D (80 males, 67 females), who were divided into 3 equal tertiles (*n* = 49) in terms of low, average and high LTL values. Among all patients with T2D: 45 (31%) suffered from DN, 109 (76%) from DPN, 43 (29%) from DFU, and 38 (26%) from DR.

Table 1 summarizes general demographic, clinical and biochemical characteristics of the patients. No significant differences were observed between the studied groups for such parameters as gender, age, duration of diabetes, age at diagnosis, BMI, WC, HbA1c, CRP, vitamin D_3_, LDL, HDL and TG levels. There were also no significant differences between the groups in the prevalence of T2D-related complications: DN, DPN, DFU and DR.

Nevertheless, significant differences were found between the shortest 1st LTL tertile and the longest 3rd LTL tertile in the levels of two well-recognized T2D risk factors TC and Non-HDL, respectively: 161.51 vs. 135.86 (mg/dL) (*p* = 0.013) and 135.86 vs. 125.08 (mg/dL) (*p* = 0.013).

However, analyses conducted in the group of T2D patients suffering from DN showed significant differences between 1st LTL tertile and 3rd LTL tertile for LDL levels 81.33 vs. 55.29 (mg/dL) (*p* = 0.023) (Table 2). Also, in the group of T2D patients with DR, significant differences were observed between LTL tertiles for such parameters as duration of diabetes 23.5 vs. 12 (years) (*p* = 0.043), HbA1c 8.75 vs. 7.4 (%) (*p* = 0.041), TC 149.33 vs. 125.24 (mg/dL) (*p* = 0.003), LDL 67.08 vs. 62.06 (mg/dL) (*p* = 0.015), Non-HDL 115.33 vs. 90.24 (mg/dL) (*p* = 0.004) and TG 151.5 vs. 112.00 (mg/dL) (*p* = 0.045) (Table 3). In contrast, no significant differences were observed in the groups of T2D patients with DPN or DFU (Appendix A).

There were no significant associations found between demographic, clinical and biochemical characteristics of the T2D patients and the LTL using Pearson correlation and Spearman’s rank correlation (Table 4 and Table 5). Likewise, there was no correlation between any of the four diabetic complications and LTL (Table 6).

Logistic regression analysis conducted on the crude model 3rd tertile showed that TC, LDL and Non-HDL levels had significantly reduced chance of having the longest LTL, respectively: OR of 0.74 (95% CI 0.60–0.92, *p* = 0.006), OR of 0.83 (95% CI 0.69–1.00, *p* = 0.047) and OR of 0.75 (95% CI 0.62–0.92, *p* = 0.005) (Appendix A). When this analysis was repeated on this model adjusted for sex and age, then conversely, TC and Non-HDL had a significant association with the longest LTL: OR of 0.71 (95% CI 0.56–0.91, *p* = 0.006) and OR of 0.73 (95% CI 0.58–0.91, *p* = 0.006), respectively (Appendix A, Figure 2). There were, however, no statistically significant associations found in any of the remaining tertiles and variables upon logistic regression analysis.

## 3. Discussion

Numerous studies have shown that telomeres play an important role in cellular aging and cell death. The literature data indicate that shortened telomeres exhibit increased mobility within the nucleus intended to result in the induction of appropriate mechanisms for their repair or elongation. It has been observed that the effect of telomeres on nuclear plasticity is mediated, among other things, by interactions between telomeres and lamina A/C and other chromatin domains, which can lead to changes in cellular nanomechanics [31,32]. LTL analysis is therefore considered as a useful biomarker of biological age [7]. The present study has attempted to verify whether LTL analysis of patients with T2D could be a useful indicator of the risk of known diabetes complications: DN, DPN, DFU and DR.

Nearly 150 Caucasian patients with T2D were enrolled for the study; the majority being men (*n* = 80). The subject data were broken down according to LTL into three equal tertile groups (*n* = 49), minimum, average and maximum LTL. On this basis, a series of comparisons and statistical analyses were performed. The analyses revealed no significant differences found between the tertile groups in regard to demographic, biochemical, nor clinical variables. However, statistically significant differences were observed for two lipid panel parameters TC and Non-HDL (*p* = 0.013). It was found that those subjects with the shortest LTL had significantly higher levels of these two variables compared with the third tertile.

When the tested group was limited to T2D patients suffering from DN, a significant difference between LTL tertiles for LDL levels (*p* = 0.023) was observed. While in the group of T2D patients with DR, significant differences between LTL tertiles were also observed for such parameters as duration of diabetes (*p* = 0.043), HbA1c (*p* = 0.041), TC (*p* = 0.003), LDL (*p* = 0.015), Non-HDL (*p* = 0.004), and TG (*p* = 0.045). No significant differences were observed in the groups of T2D patients with DPN or DFU.

In addition, a significant relationship between TC or Non-HDL and LTL was confirmed using logistic regression in the crude model as well as when adjusted for sex and age. This is quite important when considering dyslipidemia; a commonly occurring complication of T2D.

Similar results were obtained in a study conducted on a population of South Asians with T2D (*n* = 142), where male subjects demonstrated an inverse correlation between LTL with levels of TG and TC. It was postulated that the atherogenic properties of elevated TC and TG may cause increased cellular turnover and increased production of reactive oxygen species (ROS) in some cells. Moreover, it was suggested that the link between cholesterol and LTL may be secondary, where cells are driven to their maximum replicative capacity, which then translates into shortened LTL [33].

Diabetes is closely associated with dyslipidemia and a persistent state of hyperglycemia resulting from insulin resistance. Such abnormalities exacerbate ROS production and promote a pro-inflammatory environment [34]. Several studies show that oxidative stress significantly contributes to telomere shortening. Moreover, telomere repeats (triplet *G*) have been found to be exceptionally susceptible to oxidative stress-induced damage [23]. It has also been proven that inflammation significantly increases cell proliferation, which consequently leads to significant shortening of telomeres [35].

Most of the measured parameters were not normally distributed (i.e., non-parametric), which may in part explain the lack of statistically significant differences between tertiles, for parameters such as duration of diabetes, and the levels of HbA1c, CRP and TG. Another reason for the statistical insignificance may be a relatively small cohort, which has insufficient statistical power to discern any underlying relationships present. Nevertheless, some trends were found upon comparing the parameters from the 1st tertile vs. 3rd tertile, i.e., the shortest LTL vs. the longest LTL. Patients from the 3rd tertile had shorter duration of diabetes compared with the 1st tertile (median, 15 vs. 17 (years)), older age at diagnosis (mean, 51 vs. 48 (years)), smaller WC (mean, 114.63 vs. 116.96 (cm)), lower levels of HbA1c (median, 7.80 vs. 8.30 (%)) and CRP (median, 3.60 vs. 4.23 (mg/L)), along with higher level of vitamin D_3_ (mean, 22.79 vs. 19.57 (ng/mL)) and lower level of both LDL (mean, 68.59 vs. 79.67 (mg/dL)) and TG (median, 131.00 vs. 146.00 (mg/dL)). These results demonstrate that patients with long LTL had objectively more physiologically favorable parameters compared to those with short LTL.

However, it must be kept in mind that chronic conditions, such as T2D, can be prevented or treated more effectively with a lifestyle-focused approach (Lifestyle medicine), reducing their incidence and related complications [36]. The onset of T2D complications might be modified by appropriate case management, i.e., optimization of therapeutic and care strategy, focusing primarily on meeting the need for self-care and improving the patient’s overall lifestyle [37]. Furthermore, biochemical blood parameters can be effectively modified by changing lifestyle and/or diet. It has been recently reported that Mediterranean diet may affect diastolic blood pressure, HDL and glycemia [38]. Thus the relationship between LTL and blood parameters might be specific for a particular geographical region and/or lifestyle of patients. On the other hand, results of this study clearly indicate that length of LTL might be a useful prognostic marker and bring benefits for the management of chronic diseases, like Lifestyle Medicine and Healthy Aging, from both clinical and healthcare perspectives.

Many studies have been conducted on various demographic or biochemical parameters and their relationship to LTL in patients suffering from T2D. A meta-analysis of 17 publications between 1990 and 2015 showed a strong association between the prevalence of diabetes and LTL that was influenced by different factors, including geographic region where the studies had been conducted, patients’ age, type of diabetes, BMI and gender [39]. A study of 121 Pakistani patients with T2D showed a significant negative correlation between LTL and patients’ age, which was also found to be significantly influenced by hypertension, a moderate level of smoking, and then by factors with weaker influence such as gender, BMI, stress level and age at diagnosis [40]. A subsequent cross-sectional study of 90 patients with T2D (45 patients, 45 controls) showed a positive correlation between LTL and vitamin D_3_ levels, where it was observed that high HbA1c levels and lower vitamin D_3_ levels led to telomere shortening [41]. A further study on 136 subjects (34 T1D patients, 62 T2D patients and 40 controls) showed that LTL was significantly negatively correlated with age, WC, waist-to-hip ratio, and HbA1c [34]. A study on 108 Brazilian patients with T2D and 125 controls likewise observed that LTL in patients with T2D was negatively correlated with WC, duration of diabetes, fasting glucose levels, and HbA1c [29].

Interestingly, a study on Kuwaiti residents (110 T2D patients and 94 controls) showed that disease duration did not affect LTL (from 1 month to 40 years) and furthermore did not correlate with age, BMI or any glycemic parameters. It was stated that telomere shortening may be a risk factor for T2D rather than a result of the disease [6]. According to Rosa et al., the literature data indicates that LTL is a dynamic trait, which can change over time and that factors such as regular exercise and pharmacological glycemic control can increase LTL in patients with T2D up to values similar to those of people without diabetes HbA1c [29].

The present study, however, showed no statistically significant differences in patients with T2D between LTL and the incidence of common diabetic complications. Furthermore, there were no significant associations found between these factors using Pearson’s correlation and logistic regression analysis. Interestingly, a comparison of the 1st tertile vs. 3rd tertile revealed contradictory trends. Patients with the longest LTL (3rd tertile) had a decreased incidence of DPN when compared to the 1st tertile (77.60 vs. 73.50 (%)), while increasing incidences were correspondingly observed for both DFU and DR when comparing the same groups: 28.60 vs. 34.70 and 24.50 vs. 34.70 (%), respectively. It can again be concluded that the relatively small number of patients in groups may have distorted a proper picture of the underlying relationships under study.

In contrast, an Italian study (501 T2D patients and 400 controls) has demonstrated that T2D patients with additional complications had significantly shorter LTL compared to those without complications [42]. Another study on 99 subjects (50 patients with T2D, 49 controls) has subsequently shown that vascular lesions were significantly larger, and LTLs were shorter, in patients with T2D compared to controls. These findings indicate that chronic hyperglycemia in T2D patients increases protein glycation and accumulation of glycated products, which increased vascular stiffness and, consequently, led to a more rapid aging of blood vessel walls. Similar findings were also reported by Dudinskaya et al., where patients with long LTL demonstrated smaller vascular changes, as compared with healthy individuals [35].

A study on patients with either T1D or T2D revealed that a decrease of 0.1 LTL unit had increased the risk of developing chronic kidney disease by 16% [43]. A study on 691 Asian patients with T2D likewise revealed that subjects with advanced albuminuria had shorter median LTLs compared to the group with no progression, and the subjects with shorter LTL had an almost 2-fold increased risk of progression of albuminuria after accounting for traditional risk factors. LTL analysis may therefore be a useful biomarker of albuminuria progression in those patients where renal filtration is preserved, in addition to other traditional risk factors for DN such as hypertension, hyperglycemia, long duration of diabetes and dyslipidemia [23]. Furthermore, another study on 4768 subjects from Singapore (1628 T2D patients and 3140 controls) showed that shorter LTL was associated with a higher risk of chronic kidney disease in patients with T2D [44]. A study on 4085 Chinese patients with T2D showed similar outcomes, as T2D patients with the shortest LTL had a 1.8-fold higher risk of end-stage kidney disease (ESKD), as compared to the group of patients with the longest LTL. It was suggested that LTL could be a useful marker for progression of renal function and the occurrence of ESKD in T2D patients [45]. A study on telomerase activity in 90 T2D patients with or without DFU revealed that patients with DFU had a significantly higher WC, neuropathic impairment score, and lower telomerase activity [28].

A survey conducted on 120 subjects, including 36 controls, demonstrated that the groups with non-proliferative and proliferative DR had significantly lower mean LTL compared to patients without DR. It was suggested that longer LTL and poor glycemic control in patients with T2D are associated with advanced stages of DR. Findings indicate that the relative length of telomeres in patients with DR is related to the intensity of this complication/condition [46]. On the other end, a study on a South Indian population showed that DPN had no effect on LTL shortening in patients with T2D, and additionally, no strong association between LTL and the presence of T2D was observed. It was thus suggested that it is not entirely clear whether LTL shortening is a cause or a consequence of diabetes [47].

It thus appears that the precise relationship between LTL and the incidence of T2D complications still remains to be fully established. Important factors that should be taken into consideration are to have sufficient sample sizes for meaningful statistics and to account for the inherent variations in the geographical region of any given study.

In keeping with other studies, some limitations of the present study must be noted. The most important, general limitation is the somewhat limited sample size that might result in insufficient statistical power, resulting in missing some underlying relationships. Another general limitation is that in the present study, LTL was measured only in peripheral blood leukocytes, but not in diabetes target organs, such as the pancreas or liver. There was also no account taken of subjects’ lifestyle habits, such as drinking, smoking habits, or physical activity, that have been proven to affect telomere length. A methodological limitation is the method of LTL measurement. The LTL has been measured by a PCR-based assay, which does not quantify absolute telomere length and may therefore have missed differences in LTL between the study subgroups.

## 4. Materials and Methods

The study cohort consisted of 147 Caucasian patients (67 women, and 80 men) aged between 39 and 87 years (median age, 65 years), who had volunteered to take part in the study as part of their in-patient stay at the Department of Diabetes at the Institute of Rural Health in Lublin, Poland. Study participants came from the Lubelskie voivodship (provincial) region and surrounding areas. Written informed consent was obtained from each study recruit. The study protocol was approved by the local Bioethics Committee of the Institute of Rural Health in Lublin (No 13/2020). The study was conducted between 2021 and 2024, in accordance with the STROBE guidelines [48].

### 4.1. Data Collection

Information was extracted from hospital records in order to comprehensively characterize the patients. The data included information from routine clinical interviews and the results of medical examinations performed as part of hospital admission consisting of gender, age (years), height (cm) and weight (kg), BMI (kg/m^2^), waist circumference (WC) (cm), vitamin D_3_ (ng/mL), type of diabetes, duration of diabetes (years), age at diagnosis (years), glycated hemoglobin (HbA1c) (%), C-reactive protein (CRP) (mg/L), estimated glomerular filtration rate (eGFR) (mL/min/1.73 m^2^), determination of pain intensity (DPI) (VAS) and lipid profiling, consisting of total cholesterol (TC) (mg/dL), low-density lipoprotein (LDL) (mg/dL), high-density lipoprotein (HDL) (mg/dL), non-high-density lipoprotein (Non-HDL) (mg/dL) and triglycerides (TG) (mg/dL), together with the presence or absence of DN, DPN, DFU and DR.

### 4.2. Biochemical Analysis

All biochemical measurements were performed at the ALAB Medical Analysis Laboratory in Lublin, Poland.

### 4.3. DNA Extraction

Peripheral blood from patients was collected into 4.9 mL tubes coated with K_3_EDTA (Sarstedt, Nümbrecht, Germany), aliquoted into suitable volumes, and stored at −20 °C. Genomic DNA was extracted from 200 µL of whole blood using the QIAamp^®^ DNA Mini Kit (Qiagen, Hilden, Germany) according to the protocol provided by the manufacturer. The quality and quantity of isolated DNA were measured using a NanoDrop^®^ 1000 spectrophotometer (Thermo Fisher Scientific, Waltham, MA, USA). The isolated DNA was then aliquoted and stored at −20 °C for further analysis.

### 4.4. Telomere Length Measurement

LTL in DNA from patients’ peripheral blood was determined using the method of Cawthon et al. [49] modified according to O’Callaghan et al. [50]. qPCR was performed in an ABI 7500 Fast Real-Time PCR System (Applied Biosystems^®^, Thermo Fisher Scientific, Waltham, MA, USA). The standard curve quantitation method was used according to the manufacturer’s protocol (Applied Biosystems^®^; Thermo Fisher Scientific, Waltham, MA, USA). The total volume of the reaction mixture was 20 µL and consisted of 4 µL of DNA (2.5 ng/µL), 10 µL of 1X FastStart Universal SYBR Green Master (Rox) (Merck KGaA, Darmstadt, Germany), 0.2 µL primers (100 nM) (telomere forward primer 5′-CGGTTTGTTTGGGTTTGGGTTTGGGTTTGGGTTTGG GTT-3′, telomere reverse primer 5′-GGCTTGCCTTACCCTTACCCTTACCCTTACCCTT ACCCT-3′) or (36B4 forward primer 5′-CAGCAAGTGGGAAGGTGTAATCC-3′, 36B4 reverse primer 5′-CCCATTCTATCATCAACGGGTACAA-3′) and 5.6 µL of Invitrogen™ Nuclease-Free Water (not DEPC-Treated; Thermo Fisher Scientific, Waltham, MA, USA). Cycling conditions for both telomere and 36B4 products were as follows: 10 min at 95 °C, followed by 40 cycles of 95 °C for 15 s and 60 °C for 1 min. LTL measurement was represented by the T/S ratio of amplification of telomere DNA sequences (T) normalized by the single copy 36B4 gene (S) produced within each reaction and compared to a common serially diluted DNA standard curve over the 0.02–20 ng range. Samples were measured in triplicate, and the mean of the three T/S ratios was calculated.

### 4.5. Statistical Analysis

Statistical analysis was performed using MS Office Excel 2013 (Microsoft Corporation, Redmond, WA, USA), GrapPad Prism 9.5.1. (GraphPad Software Inc., Boston, MA, USA), IBM SPSS Statistics ver. 29 (SPSS Inc., IBM Corporation, Armonk, New York, NY, USA), and RStudio Desktop ver. 2024.09.1+394 (Posit Software, Boston, MA, USA). All scale variables were tested for normality of distribution using the Shapiro–Wilk test. For several variables, normality of the distributions was achieved by root transformation. For variables such as age at diagnosis, BMI, vitamin D_3_, TC, LDL and Non-HDL, it was a second-degree root, while a third-degree root was used for the HDL variable. The transformed variables were used in analyses of significance differences and correlation analyses. The use of transformed variables is relevant only for parametric tests (analysis of variance, Pearson’s correlation coefficient), while it does not affect the results of non-parametric tests (Kruskal–Wallis test, Mann–Whitney U test, Spearman’s correlation coefficient). In the tables of descriptive statistics, raw variables were included. Data were presented as median (interquartile range) or mean (standard deviation) in scale and ordinal variables. For variables with a normal distribution, analyses were examined using a one-way ANOVA test. For variables with a lack of normal distribution, the Kruskal–Wallis test was performed. For categorical variables, the Chi-square test was applied. Pearson’s and Spearman’s correlation analyses were used to detect any possible correlation between parameters. Logistic regression models were also applied to assess the risk of LTL association with some of the studied parameters. Differences were considered significant at *p* < 0.05.

## 5. Conclusions

T2D is one of the most prevalent health disorders in the world, and its incidence is predicted to continuously rise for the next few years. The chronic nature of this condition is associated with the development of many complications that can significantly impair patient health and well-being. It is thereby important to continually search for new biomarkers to predict their progression and intensity. This is also important for developing effective, targeted therapies.

Many studies indicated that measurement of LTL in lymphocytes from patients’ peripheral blood can be a useful indicator for many chronic diseases associated with aging and cell death. However, the present study has not proven any significant association between LTL and the occurrence of specific diabetic complications, such as DN, DPN, DFU and DR. Nevertheless, a significant association has been demonstrated between LTL from patients with T2D and TC or Non-HDL levels, between LTL from patients with DN and LDL levels, and also between LTL from patients with DR and duration of diabetes, HbA1c, TC, LDL, Non-HDL or TG levels. This therefore demonstrates the promising potential of LTL analysis. It appears that the biggest limitation of the present study was an insufficient number of subjects that made it difficult to observe any significant correlations, if present. Further multicenter studies on LTL and diabetes are therefore needed, but on a much larger group of patients, to address this question. Further studies should be completed on changes in telomere length over time, as well as their changes in target tissues. Taking into account lifestyle variables such as active addictions, exercise and diet. It also seems that additional valuable information would be provided by analysis of single nucleotide polymorphisms in the telomerase reverse transcriptase sequence encoded by the TERT gene.

## Figures and Tables

**Figure 1 ijms-26-00290-f001:**
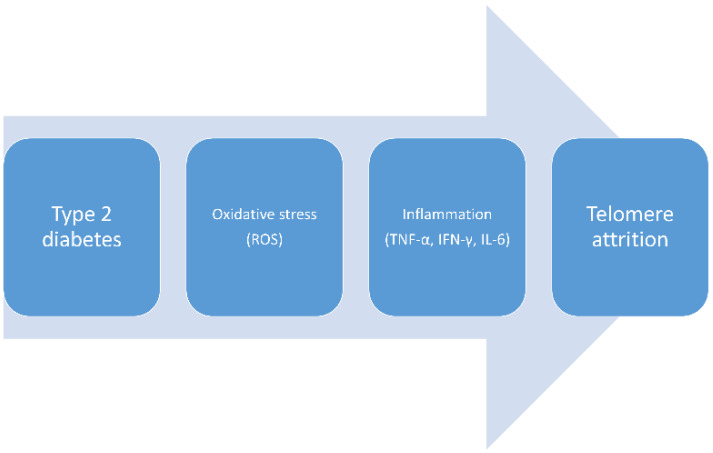
A schematic diagram showing the relationship among type 2 diabetes and telomere attrition.

**Figure 2 ijms-26-00290-f002:**
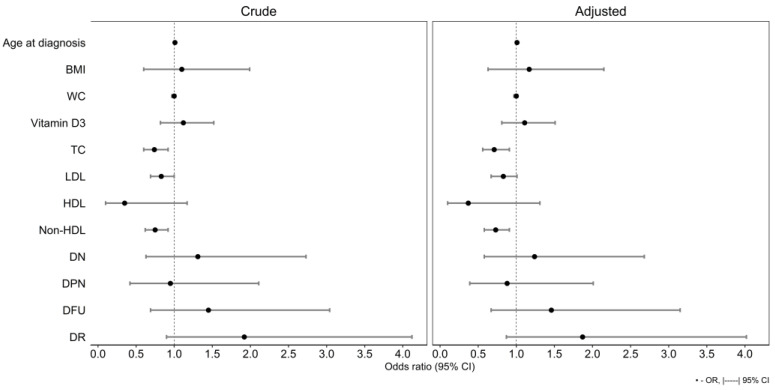
Visualization of logistic regression risk assessment for testing the association of LTL 3rd tertile with selected variables (crude and adjusted for sex and age model). LTL—leukocyte telomere length, OR—odds ratio, CI—confidence interval, BMI—body mass index, WC—waist circumference, TC—total cholesterol, LDL—low-density lipoprotein, HDL—high-density lipoprotein, Non HDL—non-high-density lipoprotein, DN—diabetic nephropathy, DPN—diabetic polyneuropathy, DFU—diabetic foot ulcer, DR—diabetic retinopathy.

**Table 1 ijms-26-00290-t001:** All Caucasian patients demographics and their clinical and biochemical characteristics broken down into tertiles according to leukocyte telomere length.

Parameters	Total	LTL Tertile			*p* Value	Test
		Tertile 1	Tertile 2	Tertile 3		
*n*	147	49	49	49		
LTL	1.04 (0.34)	0.67 (0.16)	1.04 (0.08)	1.42 (0.21)		
Men (%)	54.42	55.10	51.00	57.10	0.825	χ^2^
Age (years)	65.00 (10.00)	66.00 (10.00)	63.00 (9.00)	65.00 (10.00)	0.357	*ANOVA*
Duration of diabetes (years)	15.00 (12.00)	17.00 (15.00)	15.00 (11.00)	15.00 (16.00)	0.182	KW
Age at diagnosis (years)	49.00 (11.00)	48.00 (10.00)	49.00 (12.00)	51.00 (11.00)	0.480	*ANOVA*
BMI (kg/m^2^)	34.56 (6.91)	34.49 (6.23)	34.27 (6.38)	34.90 (8.04)	0.942	*ANOVA*
WC (cm)	114.93 (14.68)	116.96 (12.77)	113.27 (13.87)	114.63 (17.07)	0.465	*ANOVA*
HbA1c (%)	8.20 (2.50)	8.30 (2.00)	8.00 (2.90)	7.80 (2.50)	0.671	KW
CRP (mg/L)	4.36 (10.40)	4.23 (8.80)	5.72 (10.70)	3.60 (14.30)	0.965	KW
vitamin D_3_ (ng/mL)	21.93 (10.46)	19.57 (9.23)	23.42 (10.65)	22.79 (11.20)	0.157	*ANOVA*
TC (mg/dL)	152.94 (56.40)	161.51 (70.70)	161.63 (48.79)	135.86 (42.82)	**0.013**	*ANOVA*
LDL (mg/dL)	78.14 (39.87)	79.67 (41.84)	86.38 (41.48)	68.59 (34.72)	0.079	*ANOVA*
HDL (mg/dL)	37.77 (13.31)	37.53 (11.35)	38.54 (9.02)	37.27 (18.05)	0.192	*ANOVA*
Non-HDL (mg/dL)	116.23 (53.30)	125.08 (68.36)	123.38 (44.84)	100.39 (39.39)	**0.013**	*ANOVA*
TG (mg/dL)	143.50 (104.00)	146.00 (102.00)	148.50 (104.50)	131.00 (99.00)	0.182	KW
eGFR (<60 mL/min/1.73 m^2^) (%)	29.25	30.60	22.40	34.70	0.435	χ^2^
DPI (VAS)	6.00 (8.00)	7.00 (9.00)	5.00 (7.00)	6.00 (8.00)	0.125	KW
DN (%)	30.82	32.70	24.50	34.70	0.553	χ^2^
DPN (%)	75.69	77.60	71.40	73.50	0.931	χ^2^
DFU (%)	29.45	28.60	24.50	34.70	0.570	χ^2^
DR (%)	26.03	24.50	18.40	34.70	0.193	χ^2^

Data are expressed as mean (SD), median (interquartile range) or no. (percentage) for categorical variables. *p*-values compare LTL 3rd tertile (max) to LTL 1st tertile (min). Statistically significant *p*-values bolded. LTL—leukocyte telomere length, χ^2^—Chi-square test, ANOVA—one-way ANOVA test, KW—Kruskal-Wallis test, SD—standard deviation, BMI—body mass index, WC—waist circumference, HbA1c—hemoglobin A1C (glycated hemoglobin), CRP—C-reactive protein, TC—total cholesterol, LDL—low-density lipoprotein, HDL—high-density lipoprotein, Non HDL—non-high-density lipoprotein, TG—triglycerides, eGFR—estimated glomerular filtration rate, DPI—determination of pain intensity, VAS—visual analogue scale, DN—diabetic nephropathy, DPN—diabetic polyneuropathy, DFU—diabetic foot ulcer, DR—diabetic retinopathy.

**Table 2 ijms-26-00290-t002:** Caucasian patients with diabetic nephropathy demographics and their clinical and biochemical characteristics broken down into tertiles according to leukocyte telomere length.

Parameters	Total	LTL Tertile			*p* Value	Test
		Tertile 1	Tertile 2	Tertile 3		
*n*	45	16	12	17		
LTL	1.06 (0.37)	0.67 (0.16)	1.05 (0.08)	1.43 (0.20)		
Men (%)	48.90	43.80	41.70	58.80	0.580	χ^2^
Age (years)	68.76 (9.27)	68.63 (10.47)	67.42 (8.52)	69.82 (9.01)	0.795	*ANOVA*
Duration of diabetes (years)	20.00 (15.00)	23.00 (14.00)	15.00 (10.00)	20.00 (16.00)	0.267	KW
Age at diagnosis (years)	49.02 (10.43)	46.94 (10.37)	50.45 (9.61)	50.06 (11.28)	0.613	*ANOVA*
BMI (kg/m^2^)	35.38 (6.81)	35.13 (6.53)	34.58 (5.67)	36.18 (8.03)	0.845	*ANOVA*
WC (cm)	116.28 (13.72)	118.13 (12.62)	112.77 (10.96)	117.12 (16.45)	0.581	*ANOVA*
HbA1c (%)	8.20 (1.90)	8.20 (1.45)	8.15 (3.60)	7.80 (2.30)	0.803	KW
CRP (mg/L)	7.20 (11.26)	8.48 (13.18)	8.19 (7.36)	4.22 (15.79)	0.900	KW
vitamin D_3_ (ng/mL)	20.74 (10.54)	16.68 (8.03)	24.40 (11.51)	21.96 (11.22)	0.140	*ANOVA*
TC (mg/dL)	143.22 (41.70)	159.38 (44.93)	145.33 (44.45)	126.53 (31.17)	0.076	*ANOVA*
LDL (mg/dL)	67.93 (28.48)	81.33 (26.61)	69.08 (33.00)	55.29 (21.74)	**0.023**	*ANOVA*
HDL (mg/dL)	36.18 (12.07)	38.25 (11.36)	34.83 (10.88)	35.18 (13.85)	0.607	*ANOVA*
Non-HDL (mg/dL)	107.16 (37.93)	121.38 (42.99)	110.50 (37.51)	91.41 (27.99)	0.072	*ANOVA*
TG (mg/dL)	149.00 (114.00)	145.00 (110.00)	156.00 (140.50)	153.00 (115.00)	0.698	KW
eGFR (<60 mL/min/1.73 m^2^) (%)	95.60	93.80	91.70	100.00	0.511	χ^2^
DPI (VAS)	7.00 (4.00)	7.00 (8.00)	7.00 (5.00)	7.00 (3.50)	0.802	KW

Data are expressed as mean (SD), median (interquartile range) or no. (percentage) for categorical variables. *p*-values compare LTL 3rd tertile (max) to LTL 1st tertile (min). Statistically significant *p*-values bolded. LTL—leukocyte telomere length, χ^2^—Chi-square test, ANOVA—one-way ANOVA test, KW—Kruskal-Wallis test, SD—standard deviation, BMI—body mass index, WC—waist circumference, HbA1c—hemoglobin A1C (glycated hemoglobin), CRP—C-reactive protein, TC—total cholesterol, LDL—low-density lipoprotein, HDL—high-density lipoprotein, Non HDL—non-high-density lipoprotein, TG—triglycerides, eGFR—estimated glomerular filtration rate, DPI—determination of pain intensity, VAS—visual analogue scale.

**Table 3 ijms-26-00290-t003:** Caucasian patients with diabetic retinopathy demographics and their clinical and biochemical characteristics broken down into tertiles according to leukocyte telomere length.

Parameters	Total	LTL Tertile			*p* Value	Test
		Tertile 1	Tertile 2	Tertile 3		
*n*	38	12	9	17		
LTL	1.08 (0.39)	0.67 (0.12)	1.01 (0.06)	1.42 (0.30)		
Men (%)	60.50	41.70	55.60	76.50	0.158	χ^2^
Age (years)	65.87 (9.98)	67.83 (9.47)	61.89 (10.37)	66.59 (10.14)	0.381	*ANOVA*
Duration of diabetes (years)	19.00 (15.00)	23.50 (9.50)	17.00 (9.00)	12.00 (15.00)	**0.043**	KW
Age at diagnosis (years)	48.18 (9.50)	45.25 (6.47)	44.67 (9.90)	52.12 (10.06)	0.066	*ANOVA*
BMI (kg/m^2^)	32.58 (5.52)	32.98 (4.74)	35.06 (5.46)	30.98 (5.83)	0.194	*ANOVA*
WC (cm)	111.01 (12.33)	113.00 (12.47)	113.02 (7.55)	108.65 (14.34)	0.576	*ANOVA*
HbA1c (%)	8.45 (2.00)	8.75 (1.75)	9.20 (2.80)	7.40 (2.00)	**0.041**	KW
CRP (mg/L)	5.83 (17.53)	5.49 (7.98)	10.21 (13.18)	3.60 (22.10)	0.402	KW
vitamin D_3_ (ng/mL)	22.00 (10.62)	17.92 (7.91)	23.01 (9.59)	24.34 (12.36)	0.287	*ANOVA*
TC (mg/dL)	147.32 (46.30)	149.33 (45.72)	186.33 (35.73)	125.24 (38.78)	**0.003**	*ANOVA*
LDL (mg/dL)	73.84 (36.97)	67.08 (32.09)	105.11 (37.72)	62.06 (31.78)	**0.015**	*ANOVA*
HDL (mg/dL)	36.16 (9.56)	34.08 (5.43)	40.89 (7.27)	35.12 (12.16)	0.180	*ANOVA*
Non-HDL (mg/dL)	111.24 (43.78)	115.33 (43.32)	145.44 (33.28)	90.24 (38.06)	**0.004**	*ANOVA*
TG (mg/dL)	142.00 (103.00)	151.50 (86.00)	201.00 (78.00)	112.00 (108.00)	**0.045**	KW
eGFR (<60 mL/min/1.73 m^2^) (%)	39.50	41.70	44.40	35.30	0.886	χ^2^
DPI (VAS)	6.00 (4.00)	7.50 (7.50)	5.00 (3.00)	6.50 (5.50)	0.571	KW

Data are expressed as mean (SD), median (interquartile range) or no. (percentage) for categorical variables. *p*-values compare LTL 3rd tertile (max) to LTL 1st tertile (min). Statistically significant *p*-values bolded. LTL—leukocyte telomere length, χ^2^—Chi-square test, ANOVA—one-way ANOVA test, KW—Kruskal-Wallis test, SD—standard deviation, BMI—body mass index, WC—waist circumference, HbA1c—hemoglobin A1C (glycated hemoglobin), CRP—C-reactive protein, TC—total cholesterol, LDL—low-density lipoprotein, HDL—high-density lipoprotein, Non HDL—non-high-density lipoprotein, TG—triglycerides, eGFR—estimated glomerular filtration rate, DPI—determination of pain intensity, VAS—visual analogue scale.

**Table 4 ijms-26-00290-t004:** Pearson correlation between normally distributed (parametric) variables and LTL in Caucasian patients with T2D.

Variables	Pearson Correlation	Sig. (2-Tailed)
Age	0.058	0.482
Gender	−0.012	0.887
Age at diagnosis	0.117	0.164
BMI	0.011	0.899
WC	−0.040	0.630
Vitamin D_3_	0.078	0.347
TC	−0.109	0.193
LDL	−0.067	0.429
HDL	−0.067	0.422
Non-HDL	−0.118	0.159

LTL—leukocyte telomere length, BMI—body mass index, WC—waist circumference, TC—total cholesterol, LDL—low-density lipoprotein, HDL—high-density lipoprotein, Non HDL—non-high-density lipoprotein.

**Table 5 ijms-26-00290-t005:** Spearman’s rank correlation between non-normally distributed (non-parametric) variables and LTL in Caucasian patients with T2D.

Variables	Spearman’s Rho	Sig. (2-Tailed)
Duration of diabetes	−0.109	0.194
HbA1c	−0.098	0.240
CRP	0.010	0.903
TG	−0.124	0.137
DPI	−0.086	0.308

LTL—leukocyte telomere length, HbA1c—hemoglobin A1C (glycated hemoglobin), CRP—C-reactive protein, TG—triglycerides, DPI—determination of pain intensity.

**Table 6 ijms-26-00290-t006:** Pearson correlation between T2D complications and LTL.

Variables	Pearson Correlation	Sig. (2-Tailed)
DN	0.005	0.955
DPN	0.002	0.981
DFU	−0.017	0.838
DR	0.047	0.569

LTL—leukocyte telomere length, DN—diabetic nephropathy, DPN—diabetic polyneuropathy, DFU—diabetic foot ulcer, DR—diabetic retinopathy.

## Data Availability

All data are contained within the article/Appendix A. Raw data points from the study are available on request from the corresponding author.

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
