# Peer review of "Leukocyte Telomere Length as a Marker of Chronic Complications in Type 2 Diabetes Patients: A Risk Assessment Study"

_ijms, 2024, doi:10.3390/ijms26010290_

Round 1

Reviewer 1 Report

Comments and Suggestions for Authors

The manuscript titled “A risk assessment study to investigate whether leukocyte telomere length can be used as a marker of chronic complications in type 2 diabetes patients” by Sawicki, K.; et al. is a scientific work where the authors assessed the levels of leukocyte telomere length in patients with diabetes with promising results. This could act as prognosis factor for this human disease. The most relevant findings encountered in this research could serve to pave the way in the design of the next-generation of smart and customized therapies against type 2 diabetes. The manuscript is generally well-written and this is a topic of growing interest.

However, it exists some points that need to be addressed (please, see them below detailed point-by-point) to improve the scientific quality of the submitted manuscript paper before this article will be consider for its publication in the International Journal of Molecular Sciences.

1) “Diabetes poses one of the biggest global challenges for public health (…) 90% of all diagnosed cases of diabetes (…) 415 million people worldwide suffering from diabetes (…) 580 million people will have this condition by 2030” (lines 35-39). Could the authors provide quantitative data insights according to the worldwide global burdens of disability-adjusted life-years (DALYs) associated to diabetes? This will aid the potential readers to better understand the significance of this devoted research.

2) “T2D is an age-related disease influenced by both genetic and environmental factors (…) diabetic retinopathy (DR), diabetic polyneuropathy (DPN), diabetic foot ulcer (DFU) and diabetic nephropathy (DN) (…) leading to end-stage renal failure” (lines 40-57). Here, the existing therapies and strategies to fight against this type of diseases.

3) “Telomere shortening has been shown to be induced by (…) TNF-α, IFN-γ and IL-6 (…) biological ageing” (lines 66-70). A schematic representation highlighting the crosstalk among the cytokines and the disease will benefit to have a more complete overview of their action mechanisms.

4) “Studied cohort consisted of 147 patients diagnosed with T2D (80 males, 67 females) (…) Table 1 summarizes general demographic, clinical and biochemical characteristics of the patients” (lines 80-85). Table 1 should add data concerning the race of the patients since this is a key factor that can affect to the diabetes disease onset and progression. Same comment for the Table 2 (line 113), Table 3 (line 123) and Table 4 (line 137).

5) “Numerous studies have shown that telomeres play an important role in cellular aging and cell death (…) indicator of the risk of known diabetes complications: DN, DPN, DFU and DR” (lines 167-171). Here, even if I agree with these statements provided by the authors, it should be also discussed how the telomers can impact the nuclear plasticity [1] which can lead in changes of the cellular nanomechanics [2] and the subsequent disease symptoms.

[1] De Vos, W.H.; et al. Increased plasticity of the nuclear envelope and hypermobility of telomers due to the loss of A-type lamins. Biochim. Biophys. Acta 2010, 1800, 448-458. https://doi.org/10.1016/j.bbagen.2010.01.002

[2] Magazzù, A.; et al. Investigation of Soft Matter Nanomechanics by Atomic Force Microscopy and Optical Tweezers: A Comprehensive Review. Nanomaterials 2023, 13, 963. https://doi.org/10.3390/nano13060963

6) “5. Conclusions” (lines 376-394). This section perfectly remarks the most relevant outcomes found in this research and also the promising future prospectives. It would be advisable to furnish a brief statement to discuss about the future action lines to pursue the topic covered in this work.

Author Response

Manuscript ID: ijms-3378821

Response to Reviewer Comments

We thank the Reviewer for their comments. Below are our responses.

Review report 1

The manuscript titled “A risk assessment study to investigate whether leukocyte telomere length can be used as a marker of chronic complications in type 2 diabetes patients” by Sawicki, K.; et al. is a scientific work where the authors assessed the levels of leukocyte telomere length in patients with diabetes with promising results. This could act as prognosis factor for this human disease. The most relevant findings encountered in this research could serve to pave the way in the design of the next-generation of smart and customized therapies against type 2 diabetes. The manuscript is generally well-written and this is a topic of growing interest.

However, it exists some points that need to be addressed (please, see them below detailed point-by-point) to improve the scientific quality of the submitted manuscript paper before this article will be consider for its publication in the International Journal of Molecular Sciences.

1) “Diabetes poses one of the biggest global challenges for public health (…) 90% of all diagnosed cases of diabetes (…) 415 million people worldwide suffering from diabetes (…) 580 million people will have this condition by 2030” (lines 35-39). Could the authors provide quantitative data insights according to the worldwide global burdens of disability-adjusted life-years (DALYs) associated to diabetes? This will aid the potential readers to better understand the significance of this devoted research.

Response: As suggested by the Reviewer, we have included the relevant information in the indicated passage in the introduction:

…„The overall burden of disease is assessed using the disability-adjusted life year (DALY), a time-based measure that combines years of life lost due to premature mortality (YLLs) and years of life lost due to time lived in states of less than full health, or years of healthy life lost due to disability (YLDs). One DALY represents the loss of the equivalent of one year of full health. In 2021, 58·9 million DALYs or 76·5% of DALYs due to T2D were attributable to risk factors. Of the 16 risk factors high body mass index (BMI) was the primary risk factor for T2D worldwide, accounting for 52·2% of global T2D DALYs [4].”…

2) “T2D is an age-related disease influenced by both genetic and environmental factors (…) diabetic retinopathy (DR), diabetic polyneuropathy (DPN), diabetic foot ulcer (DFU) and diabetic nephropathy (DN) (…) leading to end-stage renal failure” (lines 40-57). Here, the existing therapies and strategies to fight against this type of diseases.

Response: As suggested by the Reviewer, we have included the relevant information in the indicated passage in the introduction:

…“Standard therapy and management regimens for T2D patients mainly include stabilizing blood glucose, blood pressure, and serum cholesterol levels. Depending on the incidence of additional complications in T2D patients, dedicated treatment protocols are used. For DR, most patients may require frequent anti- vascular endothelial growth factor (anti-VEGF) intravitreal injections, and for patients with proliferative DR, panretinal laser photocoagulation has proven effective in reducing the risk of vision loss [19]. In the case of DPN treatment, various medications are generally used to alleviate symptoms (including chronic pain) and therapeutic drugs targeting the pathogenesis. Non-pharmacological therapies are also used to supplement drug treatment. These include psychological support, acupuncture, physiotherapy and transcutaneous electrical nerve or muscle stimulation [20]. DFU therapy mainly focuses on treatments including surgical debridement of the wound, reducing the pressure exerted by body weight on the ulcer, and treating lower extremity ischemia and infection of the foot. In addition, therapy may include taking medications to accelerate wound healing and targeted oral antibiotics for localized osteomyelitis [21]. The therapeutic scheme for DN includes tight glycemic control, blood pressure control with renin-angiotensin-aldosterone system (RAAS) inhibitors, lipid-lowering agents, weight loss, and protein restriction. In addition, prevention of DN progression uses, vitamin D receptor activators, incretin-related drugs, and inflammation-targeted therapies [22].”…

3) “Telomere shortening has been shown to be induced by (…) TNF-α, IFN-γ and IL-6 (…) biological ageing” (lines 66-70). A schematic representation highlighting the crosstalk among the cytokines and the disease will benefit to have a more complete overview of their action mechanisms.

Response: In accordance with the reviewer's suggestions, we have included a corresponding simplified relationship diagram (as Figure 1) in the indicated section of the introduction.

4) “Studied cohort consisted of 147 patients diagnosed with T2D (80 males, 67 females) (…) Table 1 summarizes general demographic, clinical and biochemical characteristics of the patients” (lines 80-85). Table 1 should add data concerning the race of the patients since this is a key factor that can affect to the diabetes disease onset and progression. Same comment for the Table 2 (line 113), Table 3 (line 123) and Table 4 (line 137).

Response: As suggested by the Reviewer, we have included information about the Caucasian race of the patients participating in this study in all table descriptions as well as in the description of the patients in the Materials and Methods section and in the Discussion.

5) “Numerous studies have shown that telomeres play an important role in cellular aging and cell death (…) indicator of the risk of known diabetes complications: DN, DPN, DFU and DR” (lines 167-171). Here, even if I agree with these statements provided by the authors, it should be also discussed how the telomers can impact the nuclear plasticity [1] which can lead in changes of the cellular nanomechanics [2] and the subsequent disease symptoms.

[1] De Vos, W.H.; et al. Increased plasticity of the nuclear envelope and hypermobility of telomers due to the loss of A-type lamins. Biochim. Biophys. Acta 2010, 1800, 448-458. https://doi.org/10.1016/j.bbagen.2010.01.002

[2] Magazzù, A.; et al. Investigation of Soft Matter Nanomechanics by Atomic Force Microscopy and Optical Tweezers: A Comprehensive Review. Nanomaterials 2023, 13, 963. https://doi.org/10.3390/nano13060963

Response: In accordance with the Reviewer suggestion, we have included the relevant passage in the Discussion section:

…„Literature data indicate that shortened telomeres exhibit increased mobility within the nucleus intended to result in the induction of appropriate mechanisms for their repair or elongation. It has been observed that the effect of telomeres on nuclear plasticity is mediated, among other things, by interactions between telomeres and lamina A/C and other chromatin domains, which can lead to changes in cellular nanomechanics [31,32].”…

6) “5. Conclusions” (lines 376-394). This section perfectly remarks the most relevant outcomes found in this research and also the promising future prospectives. It would be advisable to furnish a brief statement to discuss about the future action lines to pursue the topic covered in this work.

Response: In accordance with the reviewer suggestions, we have included the relevant passage in the conclusion:

…”Further multicenter studies on LTL and diabetes are therefore needed, but on a much larger group of patients, to address this question. Further studies should be completed on changes in telomere length over time, as well as their changes in target tissues. Taking into account lifestyle variables such as active addictions, exercise and diet. It also seems that additional valuable information would be provided by analysis of single nucleotide polymorphisms in the telomerase reverse transcriptase sequence encoded by the TERT gene.”…

We thank the Reviewer for the review as well as for taking the time to read our manuscript.

On behalf of all authors, Krzysztof Sawicki, PhD

Reviewer 2 Report

Comments and Suggestions for Authors

The study investigates if leukocyte telomere length (LTL) can be a biomarker for chronic complications in Type 2 Diabetes (T2D). Involving 147 T2D patients, LTL was measured through PCR and its links to complications like diabetic retinopathy, nephropathy, polyneuropathy, and foot ulcers were assessed. No direct link was found between LTL and these complications, but associations with total cholesterol (TC) and non-HDL cholesterol were noted.

Strengths:

Innovative Focus: Exploring LTL as a biomarker for T2D complications aligns with precision medicine.

Comprehensive Methodology: Utilizes well-documented PCR-based assays for LTL measurement.

Clinical Relevance: Links LTL with dyslipidemia markers (TC and non-HDL), hinting at its metabolic relevance.

Ethical Standards: Follows ethical protocols, including informed consent and ethics committee approval.

Suggestions for Improvement:

Expand Cohort Size: Larger, multi-center studies to enhance statistical power and generalizability.

Incorporate longitudinal data: Tracking LTL changes over time to better understand disease progression.

Include lifestyle variables: Adjust for confounders like smoking, exercise, and diet for a clearer LTL role.

Broaden Analysis Techniques: To validate findings, use alternative methods like telomere-specific fluorescent in situ hybridization.

Tissue-Specific Investigations: Examine telomere dynamics in organs affected by T2D, like the pancreas or kidneys, and link with present findings.

Author Response

Manuscript ID: ijms-3378821

Response to Reviewer Comments

We thank the Reviewer for their comments. Below are our responses.

Reviewer report 2

The study investigates if leukocyte telomere length (LTL) can be a biomarker for chronic complications in Type 2 Diabetes (T2D). Involving 147 T2D patients, LTL was measured through PCR and its links to complications like diabetic retinopathy, nephropathy, polyneuropathy, and foot ulcers were assessed. No direct link was found between LTL and these complications, but associations with total cholesterol (TC) and non-HDL cholesterol were noted.

Strengths:

Innovative Focus: Exploring LTL as a biomarker for T2D complications aligns with precision medicine.

Comprehensive Methodology: Utilizes well-documented PCR-based assays for LTL measurement.

Clinical Relevance: Links LTL with dyslipidemia markers (TC and non-HDL), hinting at its metabolic relevance.

Ethical Standards: Follows ethical protocols, including informed consent and ethics committee approval.

Suggestions for Improvement:

Expand Cohort Size: Larger, multi-center studies to enhance statistical power and generalizability.

Incorporate longitudinal data: Tracking LTL changes over time to better understand disease progression.

Include lifestyle variables: Adjust for confounders like smoking, exercise, and diet for a clearer LTL role.

Broaden Analysis Techniques: To validate findings, use alternative methods like telomere-specific fluorescent in situ hybridization.

Tissue-Specific Investigations: Examine telomere dynamics in organs affected by T2D, like the pancreas or kidneys, and link with present findings.

Response: The authors agree with the reviewer's suggestions. However, it is not possible to conduct additional research at this stage due to the termination of project funding. Nevertheless, encouraged by the reviewer, we are planning a new research project on the analysis of single nucleotide polymorphisms in the telomerase reverse transcriptase sequence encoded by the TERT gene in T2D patients with complications in which the above suggestions will be taken into account.

We thank the Reviewer for the review as well as for taking the time to read our manuscript.

On behalf of all authors, Krzysztof Sawicki, PhD

Reviewer 3 Report

Comments and Suggestions for Authors

The manuscript offers a comprehensive and well-structured analysis of the potential link between leukocyte telomere length (LTL) and chronic complications in type 2 diabetes (T2D) patients. Below are positive comments that highlight the strengths of the study.

  1. Could the relatively small sample size have limited the statistical power to detect significant associations between LTL and the complications of T2D?
  2. Why was LTL in peripheral blood leukocytes chosen as the biomarker rather than measuring telomere length in diabetes-specific target tissues like the pancreas or kidneys?
  3. Could the use of a PCR-based method to measure relative telomere length (rather than absolute length) have influenced the study results? How does this method compare with other available techniques for LTL measurement?
  1. Significant associations were observed between LTL and certain lipid parameters (e.g., TC, Non-HDL) but not with the complications themselves. Could these lipid parameters be mediating factors rather than direct indicators of T2D complications?
  2. The manuscript notes trends (e.g., shorter LTL associated with higher HbA1c and lower vitamin D3 levels) despite a lack of statistical significance. Could these trends reflect a biologically relevant relationship that warrants further investigation?
  1. How do the findings of this study compare with other studies that have reported significant correlations between LTL and T2D complications, such as nephropathy or retinopathy? Could geographic or demographic differences in study populations explain the discrepancies?

Author Response

Manuscript ID: ijms-3378821

Response to Reviewer Comments

We thank the Reviewer for their comments. Below are our responses.

Reviewer report 3

The manuscript offers a comprehensive and well-structured analysis of the potential link between leukocyte telomere length (LTL) and chronic complications in type 2 diabetes (T2D) patients. Below are positive comments that highlight the strengths of the study.

  1. Could the relatively small sample size have limited the statistical power to detect significant associations between LTL and the complications of T2D?
  2. Why was LTL in peripheral blood leukocytes chosen as the biomarker rather than measuring telomere length in diabetes-specific target tissues like the pancreas or kidneys?
  3. Could the use of a PCR-based method to measure relative telomere length (rather than absolute length) have influenced the study results? How does this method compare with other available techniques for LTL measurement?
  1. Significant associations were observed between LTL and certain lipid parameters (e.g., TC, Non-HDL) but not with the complications themselves. Could these lipid parameters be mediating factors rather than direct indicators of T2D complications?
  2. The manuscript notes trends (e.g., shorter LTL associated with higher HbA1c and lower vitamin D3 levels) despite a lack of statistical significance. Could these trends reflect a biologically relevant relationship that warrants further investigation?
  1. How do the findings of this study compare with other studies that have reported significant correlations between LTL and T2D complications, such as nephropathy or retinopathy? Could geographic or demographic differences in study populations explain the discrepancies?

Response: We thank the Reviewer for their positive comments.

Ad 1. It is not possible to conduct additional research at this stage due to the termination of project funding. However, analysis of LTL and some studied parameters in T2D patients’ leukocytes demonstrates a promising potential as a marker in predicting the onset of complications in T2D. This could also help in establishing an effective treatment strategy or even prevent and delay the onset of these severe complications. Further studies are necessary, but on a much larger group of patients.

Ad 2. Obtaining target tissues like the pancreas or kidneys from patients is very difficult. I the case of diabetes patients it was impossible.

Ad 3. In our opinion, this had no impact. This is a commonly used technique for telomere analysis.

Ad 4. It is possible. More research is needed in this area.

Ad 5. Yes, and it definitely needs further research.

Ad 6. All the research we found in this area was cited in the discussion. Race matters for sure. More research is needed in this area.

We thank the Reviewer for the review as well as for taking the time to read our manuscript.

On behalf of all authors, Krzysztof Sawicki, PhD

Reviewer 4 Report

Comments and Suggestions for Authors

The T2D is one of the most spread medical condition around the world. More or less around 5% of humans are suffering from it. Quite some factors are related to it such as the age, the genetics and the environment too. The complications are various from person to person. The scientific interest is still focused on finding the right, or maybe the best approach to stop this condition or maybe to controll it easier. This topic is not new and will not be new for a while. It is still hot with other words. Studies have suggested that the telomers can be used like markers of the biological ageing. The topic of the current work is to prove if the leukocyte telomer lenght can be used as marker in predicting the onset of complications for such patients. The paper is well organized on sections, they are easy to read and understand. No english issue found too.

The telomers role is on the cellular aging and death. They can be used as biomarkers of biological age.  Nearly 150 patients were enrolled in the study. The majority was formed by man (80). The results are checked against the specific literature. As, somehow concluded, the relation between the LTL and the T3D is not that clear as expected, a lot of working still should be done.

The reference list support the findings.  No issue or observations to be done.

Author Response

Manuscript ID: ijms-3378821

Response to Reviewer Comments

We thank the Reviewer for their comments. Below are our responses.

Reviewer report 4

The T2D is one of the most spread medical condition around the world. More or less around 5% of humans are suffering from it. Quite some factors are related to it such as the age, the genetics and the environment too. The complications are various from person to person. The scientific interest is still focused on finding the right, or maybe the best approach to stop this condition or maybe to controll it easier. This topic is not new and will not be new for a while. It is still hot with other words. Studies have suggested that the telomers can be used like markers of the biological ageing. The topic of the current work is to prove if the leukocyte telomer lenght can be used as marker in predicting the onset of complications for such patients. The paper is well organized on sections, they are easy to read and understand. No english issue found too.

The telomers role is on the cellular aging and death. They can be used as biomarkers of biological age.  Nearly 150 patients were enrolled in the study. The majority was formed by man (80). The results are checked against the specific literature. As, somehow concluded, the relation between the LTL and the T3D is not that clear as expected, a lot of working still should be done.

The reference list support the findings.  No issue or observations to be done.

Response: We thank the Reviewer for the review as well as for taking the time to read our manuscript.

On behalf of all authors, Krzysztof Sawicki, PhD

Reviewer 5 Report

Comments and Suggestions for Authors

Dear Authors,

the comments in the annex file

Author Response

Manuscript ID: ijms-3378821

Response to Reviewer Comments

We thank the Reviewer for their comments. Below are our responses.

Reviewer report 5

First of all, I would like to express my sincere thanks for giving me the opportunity to contribute my opinion in evaluating your manuscript. I found the topic of the study extremely interesting and highly relevant to the field in which we work. The research presents numerous useful and promising insights, which could lead to significant advancements in our sector. However, after a thorough review, I believe that there are certain aspects that need to be improved and clarified to fully enhance the value of the proposed work. Below, I outline the main areas that could benefit from further exploration and revision.

Title: I suggest making the title clearer, possibly by specifying the type of study conducted and reducing the number of words that make it overly complex.

Response: The title was changed, to be more concise, but keeping the STROBE requirements.

Is: Leukocyte telomere length as a marker of chronic complications in type 2 diabetes patients: a risk assessment study

Was: A risk assessment study to investigate whether leukocyte telomere length can be used as a marker of chronic complications in type 2 diabetes patients

Abstract: The abstract lacks a true interpretation in the conclusions regarding the clinical implications of the phenomenon studied, along with any suggestions or specific proposals for the relevant scientific community.

Response: The abstract was modified according to the reviewer's suggestions also taking into account the journal's restrictions on the length of the abstract (limit of about 200 words):

Is: (…) No significant associations were found between LTL in T2D patients and the prevalence of common T2D complications. Nevertheless, a significant association was demonstrated between LTL and some markers of dyslipidemia, including in T2D patients with either diabetic nephropathy or retinopathy. Therefore analysis of LTL in T2D patients’ leukocytes demonstrates a promising potential as a marker in predicting the onset of complications in T2D. This could also help in establishing an effective treatment strategy or even prevent and delay the onset of these severe complications.

Was: (…) No significant associations were found between LTL in T2D patients and the prevalence of common T2D complications. Nevertheless, a significant association was demonstrated between LTL and some studied parameters. Further studies are necessary, but on a much larger group of patients.

Keywords: I recommend limiting the keywords to 4 or 5, focusing more on the main topics of the study. The current keywords are too sensitive.

Response: The number of keywords was reduced.

Is: type 2 diabetes; telomere length shortening; diabetic chronic complication

Was: type 2 diabetes; leukocyte telomere length; diabetic foot ulcer; diabetic nephropathy; diabetic polyneuropathy; diabetic retinopathy

Editing: The use of acronyms should be consistent throughout the manuscript (e.g., T2D).

Response: The acronyms were carefully checked.

Aims: The aims are not structured clearly or concisely. I suggest being more specific and dividing them into primary and secondary aims.

Response: The aims of the study were restructured and rephrased

Is: The purpose of this study is to determine if leukocyte telomere length (LTL) from T2D patients, can be a useful marker in predicting the onset of complications. A secondary aim of this study was to find any correlation between LTL and patients’ biochemical blood parameters that can be useful in prediction of the onset of disease. This could also help in establishing an effective treatment strategy or even prevent and delay the onset of these severe complications.”

Was: The purpose of this study is to determine if leukocyte telomere length (LTL), from T2D patients, can be a useful marker in predicting the onset of complications in such diabetic patients. This could also help in establishing an effective treatment strategy or even prevent and delay the onset of these severe complications.

Introduction: The epidemiological aspect should be addressed more generally (global context) before narrowing down to the specific setting. This would also make the manuscript more attractive to a wider audience. Additionally, updating the reference bibliography with more current data would strengthen the introduction.

Response: As suggested by the reviewer, we have included the relevant information in the introduction:

…” The overall burden of disease is assessed using the disability-adjusted life year (DALY), a time-based measure that combines years of life lost due to premature mortality (YLLs) and years of life lost due to time lived in states of less than full health, or years of healthy life lost due to disability (YLDs). One DALY represents the loss of the equivalent of one year of full health. In 2021, 58·9 million DALYs or 76·5% of DALYs due to T2D were attributable to risk factors. Of the 16 risk factors high body mass index (BMI) was the primary risk factor for T2D worldwide, accounting for 52·2% of global T2D DALYs [4].” …

…” Standard therapy and management regimens for T2D patients mainly include stabilizing blood glucose, blood pressure, and serum cholesterol levels. Depending on the incidence of additional complications in T2D patients, dedicated treatment protocols are used. For DR, most patients may require frequent anti- vascular endothelial growth factor (anti-VEGF) intravitreal injections, and for patients with proliferative DR, panretinal laser photocoagulation has proven effective in reducing the risk of vision loss [19]. In the case of DPN treatment, various medications are generally used to alleviate symptoms (including chronic pain) and therapeutic drugs targeting the pathogenesis. Non-pharmacological therapies are also used to supplement drug treatment. These include psychological support, acupuncture, physiotherapy and transcutaneous electrical nerve or muscle stimulation [20]. DFU therapy mainly focuses on treatments including surgical debridement of the wound, reducing the pressure exerted by body weight on the ulcer, and treating lower extremity ischemia and infection of the foot. In addition, therapy may include taking medications to accelerate wound healing and targeted oral antibiotics for localized osteomyelitis [21]. The therapeutic scheme for DN includes tight glycemic control, blood pressure control with renin-angiotensin-aldosterone system (RAAS) inhibitors, lipid-lowering agents, weight loss, and protein restriction. In addition, prevention of DN progression uses, vitamin D receptor activators, incretin-related drugs, and inflammation-targeted therapies [22].”…

Methods: This section deserves the most attention. I recommend using a reporting standard suitable for the study conducted, which is currently lacking. A clear declaration would also help the authors provide all necessary information for readers to fully understand the study. For example, I suggest adding the type of reporting used in the text, along with the relevant bibliographic reference for reporting guidelines, such as STROBE (“Strengthening the Reporting of Observational Studies in Epidemiology (STROBE): explanation and elaboration”, PLoS Med. 2007 Oct 16;4(10):e297. doi: 10.1371/journal.pmed.0040297, and “The Strengthening the Reporting of Observational Studies in Epidemiology (STROBE) statement: guidelines for reporting observational studies”. J Clin Epidemiol 2008 Apr;61(4):344-9. doi: 10.1016/j.jclinepi.2007.11.008).

Response: As suggested by the reviewer, we have included the relevant information. The study was conducted … in accordance with the STROBE guidelines [48].

Results: Clear and well-structured.

Discussion: This is a strong point, but it could certainly be enhanced with clinical references, especially concerning chronic diseases and diabetes, which are closely related to the research. The discussion could potentially be concluded by emphasizing the significant influence of the studied topic, particularly in terms of benefits for the management of chronic diseases like Lifestyle Medicine and Healthy Aging, from both clinical and healthcare perspectives. I suggest referring to relevant literature (“Lifestyle Medicine Case Manager Nurses for Type Two Diabetes Patients: An Overview of a Job Description Framework—A Narrative Review”, https://doi.org/10.3390/diabetology5040029, “Lifestyle (Medicine) and Healthy Aging”, doi: 10.1016/j.cger.2020.06.007, and Mediterranean Diet (MedDiet) and Lifestyle Medicine (LM) for support and care of patients with type II diabetes in the COVID-19 era: a cross-observational study. Acta Biomed. 2023 Aug 30;94(S3):e2023189. doi: 10.23750/abm.v94iS3.14406), which could easily align with your discussion.

Response: New paragraph regarding implications of Lifestyle Medicine in described topic was added and importance of the results for LM was emphased. References proposed by the Reviewer were included.

…”However, it must be kept in mind that chronic conditions, such as T2D, can be prevented or treated more effectively with a lifestyle-focused approach (Lifestyle medicine), reducing their incidence and related complications [36]. The onset of T2D complications might be modified by appropriate case management, i.e optimalization of therapeutic and care strategy, focusing primarily on meeting the need for self-care and improving the patient’s overall lifestyle [37]. Furthermore, biochemical blood parameters can be effectively modified by changing lifestyle and/or diet. It has been recently reported that mediterranean diet may affect diastolic blood pressure, HDL and glycemia [38]. Thus the relationship between LTL and blood parameters might be specific for a particular geographical region and/or lifestyle of patient. On the other hand, results of this study clearly indicate that length of LTL might be a useful prognostic marker and bring a benefits for the management of chronic diseases, like Lifestyle Medicine and Healthy Aging, from both clinical and healthcare perspectives.”….

Limitations: In my humble opinion, the limitations should be expanded and structured into a specific section. Currently, they are too general and do not provide much insight into the work presented.

Response: Limitations of the study were restructured and rephrased.

…“In keeping with other studies, some limitations of the presented must be noted. The most important, general limitation is somewhat limited sample size that might result in an insufficient statistical power resulting in missing some underlying relationships. Another general limitation is that in the presented study LTL were measured only in peripheral blood leukocytes, but not in diabetes target organs, such as the pancreas or liver. There was also no account taken of subjects’ lifestyle habits, such as drinking, smoking habits nor physical activity, that have been proven to affect telomere length. A methodological limitation is a method of LTL measurement. The LTL have been measured by a PCR-based assay, which does not quantify absolute telomere length and may therefore have missed differences in LTL between the study subgroups.”…

Bibliography: Good. Just needs to be expanded based on the suggested points.

Response: The bibliography has been supplemented with relevant publications.

In conclusion, the manuscript presents highly interesting scientific results but requires a series of methodological and structural improvements to enhance its overall quality. My advice is to proceed with revisions on the indicated points before publication, as, with the appropriate modifications, the manuscript could represent a significant contribution to the relevant scientific literature.

Response: We thank the Reviewer for the review as well as for taking the time to read our manuscript.

On behalf of all authors, Krzysztof Sawicki, PhD